# Realities, Challenges and Benefits of Antimicrobial Stewardship in Dairy Practice in the United States

**DOI:** 10.3390/microorganisms10081626

**Published:** 2022-08-11

**Authors:** Pamela L. Ruegg

**Affiliations:** College of Veterinary Medicine, Michigan State University, East Lansing, MI 48824, USA; plruegg@msu.edu

**Keywords:** antimicrobial stewardship, antibiotics, dairy, treatment, antimicrobial resistance

## Abstract

The use of antimicrobials for the treatment of food-producing animals is increasingly scrutinized and regulated based on concerns about maintaining the efficacy of antimicrobials used to treat important human diseases. Consumers are skeptical about the use of antibiotics in dairy cows, while dairy producers and veterinarians demonstrate ambivalence about maintaining animal welfare with reduced antimicrobial usage. Antimicrobial stewardship refers to proactive actions taken to preserve the efficacy of antimicrobials and emphasizes the prevention of bacterial diseases and use of evidence-based treatment protocols. The ability to broadly implement antimicrobial stewardship in the dairy industry is based on the recognition of appropriate antimicrobial usage as well as an understanding of the benefits of participating in such programs. The most common reason for the use of antimicrobials on dairy farms is the intramammary treatment of cows affected with clinical mastitis or at dry off. Based on national sales data, intramammary treatments comprise < 1% of overall antimicrobial use for food-producing animals, but a large proportion of that usage is a third-generation cephalosporin, which is classified as a highest-priority, critically important antimicrobial. Opportunities exist to improve the use of antimicrobials in dairy practice. While there are barriers to the increased adoption of antimicrobial stewardship principles, the structured nature of dairy practice and existing emphasis on disease prevention provides an opportunity to easily integrate principles of antimicrobial stewardship into daily veterinary practice. The purpose of this paper is to define elements of antimicrobial stewardship in dairy practice and discuss the challenges and potential benefits associated with these concepts.

## 1. Introduction

Antibiotics have been used for the treatment of bacterial diseases for more than 100 years, but bacteria continue to cause serious illness and deaths in humans and animals throughout the world. Effective treatments for bacterial diseases are based on the use of antimicrobials but the efficacy of commonly available treatments is threatened by the emergence of antimicrobial resistance. While some bacterial genera are intrinsically resistant to certain classes of antibiotics, the use of antibiotics can select for growth of resistant strains of previously susceptible bacteria. These bacteria can be disseminated in environments and create serious health risks for people and animals who may become infected with organisms that cannot be treated using standard therapies. The use of antimicrobials in human medicine contributes greatly to selection for resistance but antimicrobial usage (AMU) in animal agriculture is also a risk factor for the selection of resistant bacterial strains [1,2]. Many of the same antimicrobial classes are used in human and veterinary medicine and veterinarians are increasingly encouraged to limit AMU to maintain animal health and wellbeing. The potential impact of restricting AMU in food animals on the development of resistance is not well defined. A systematic review summarizing the effects of interventions to reduce AMU in food-animals concluded that restrictions on AMU are associated with a reduced or equal presence of resistance genes in bacteria but that the effects were not consistent [3].

The inability to identify obvious benefits for dairy farmers who reduce AMU creates challenges for veterinarians. On dairy farms, antibiotics are used to treat bacterial diseases (such as mastitis, metritis and pneumonia) that reduce animal welfare and production efficiency [4]. Consumer concerns have resulted in increased scrutiny of AMU in dairy cattle. A survey of 1000 consumers recently documented considerable concern about risks to personal health based on antibiotic usage on dairy farms. More than 90% of surveyed consumers felt that AMU in dairy cows posed some level of threat to their personal health and 25% perceived a “high threat” to human health [5]. Consumer concerns contrast with opinions of N. American dairy producers who generally believe that they are already using antimicrobials appropriately and that limitations on AMU would threaten animal health [6,7,8]. These contrasting viewpoints create challenges for veterinarians who are tasked with providing oversight of medically important antimicrobials [9] as it is difficult to convince farmers to change practices that they believe are appropriate. Mechanisms for the oversight of AMU have been formally defined and systematic, proactive actions taken by veterinarians to preserve effectiveness and the availability of antimicrobial drugs are referred to as “antimicrobial stewardship” [10]. Antimicrobial stewardship (AMS) emphasizes disease prevention and evidence-based therapeutic decisions, both of which are integral parts of routine veterinary services on dairy farms. The framework of an antimicrobial stewardship program is a potential mechanism for veterinarians to increase engagement with clients. The purpose of this paper is to define elements of AMS programs for dairy practice and discuss the challenges and potential benefits associated with AMS concepts. A systematic review of the literature was not performed, and studies were included based on their relevance to the topic of AMS on dairy farms with emphasis on the N. American dairy industry.

## 2. Occurrence of Disease and Use of Antimicrobials on Dairy Farms

On most farms, most dairy cattle are healthy and, as a proportion of the herd, relatively few cows receive antibiotics each day. Clinical mastitis is consistently the most commonly reported disease of dairy cattle, affecting at least 25% of cows each lactation period [11,12] and the high prevalence of subclinical mastitis at the end of lactation is the basis for comprehensive administration of intramammary antibiotics at dry off [13]. Other bacterial diseases that are commonly treated with antimicrobials include metritis (7–11% of cows), retained placenta (5–8% of cows), and respiratory disease (3% of cows). Among preweaned calves, bacterial diarrhea and respiratory disease are frequent disorders that are treated using antimicrobials; however, the overall mass of antimicrobials used in calves is less due to their smaller body size [14]. Based on disease incidence and treatment frequency, efforts to improve the quality of AMU on dairy farms need to be focused on improving treatments of bovine mastitis and metritis, while efforts to reduce AMU should be focused on the prevention of these diseases. In the U.S., relatively few antimicrobials are approved for use in lactating dairy cows (defined by the FDA as all cattle ≥ 20 months of age; Table 1), but multiple products are approved for use in non-lactating cattle (animals < 20 months of age). The selection of antimicrobials for systemic use in lactating cattle is limited to five approved compounds, three of which are β-lactams.

The FDA produces an annual summary of sales of medically important antimicrobials for use in food-animals [16], and products distributed for use in cattle account for 41% of the total, with tetracyclines contributing the greatest mass. While the annual summary does not distinguish between sales for use in dairy or beef cattle, products approved for injectable and intramammary (which is almost exclusively dairy) administration account for 6% and <1% of total product sales, respectively. While the proportion of overall AMU in the U.S. that can be attributed to dairy cows is relatively low, there are opportunities to improve. On U.S. dairy farms, intramammary ceftiofur is the most administered antimicrobial used to treat mastitis [11,14,17]. Ceftiofur was the only antimicrobial used on all 40 larger dairy farms enrolled in a recent study and accounted for about half of all AMU in cows [14]. While classifications of the relative importance of antimicrobials for human health vary among agencies [18], ceftiofur is classified by the World Health Organization as a “highest priority, critically important antimicrobial” [15]. As a third-generation cephalosporin that is similar to ceftriaxone (a drug used to treat several serious Gram-negative bacterial infections in humans), there are concerns about the selection and dissemination of bacteria that produce extended-spectrum beta-lactamases [19,20,21].

Several countries have restricted or enacted voluntary reductions in the usage of third-generation cephalosporins without apparent negative impacts on animal health. In the Netherlands, the use of critically important antimicrobials is allowed only after diagnostic and susceptibility testingdemonstrate that there are no other alternatives [22]. Consequently, the use of third-generation cephalosporins in the dairy sector has been reduced to essentially zero without negative impacts on animal health [23]. While all ceftiofur formulations require a prescription and only limited extra-label usage is allowed, the availability of five approved formulations (three systemic and two intramammary) and the lack of a withholding period for milk (when given systemically) have resulted in strong producer preferences for this compound. Ensuring the judicious usage of ceftiofur and other critically important antimicrobials (used in calves) should be an important priority of dairy veterinarians engaged in AMS.

Veterinarians in N. America have the opportunity to voluntarily reduce usage of ceftiofur by modifying the treatment protocols for mastitis and metritis. Mastitis is the most common bacterial disease of dairy cows and modest changes in the treatment protocols for clinical mastitis and at dry-off for subclinical mastitis can result in considerable reductions in AMU without compromising animal health. On many dairy farms, clinical mastitis is treated symptomatically without knowledge of its etiology, and many of these treatments are not necessary [24,25,26]. At least 85% of cases of clinical mastitis on most farms are non-severe when they are detected [27]; thus, immediate therapy is not required. Reducing the use of ceftiofur is very feasible as clinical trials have demonstrated that antimicrobial treatment in most cases of non-severe clinical mastitis that are culture-negative or caused by Gram-negative pathogens is not necessary [28,29]. Additionally, as compared to alternative antimicrobials, there is no indication of the superior efficacy of ceftiofur for the treatment of mastitis caused by Gram-positive bacteria; thus, efficacious alternatives are available [25,30]. A comprehensive network meta-analysis concluded that critically important antimicrobials are not necessary for treating non-severe bovine mastitis [31]. Reducing the duration of intramammary ceftiofur treatments is another simple mechanism to reduce AMU. The FDA-approved label for intramammary ceftiofur includes a flexible duration ranging from 2 to 8 days. While most producers treat until clinical signs have disappeared (about 5 days) [32], the costs of a longer-duration treatment exceeds its benefits [33] and clinical trials have not demonstrated improved outcomes based on longer durations [30,34]. One simple step to refine AMU is to reduce the duration of intramammary treatments from 5 days to 3 days. The adoption of selective treatment strategies for dry cows [35,36] and culture-guided treatments for non-severe clinical mastitis that use antimicrobials primarily to treat Gram-positive infections [24,25,26] can reduce AMU in adult cows by about 50%. There is evidence that dairy producers are gradually adopting selective treatments of mastitis as sales of intramammary antimicrobials have declined [16] and a recent observational study reported that 36% of about 14,000 cases of clinical mastitis recorded on 37 large dairy farms were managed without receiving antimicrobials [12]. These reductions are an indication that many dairy veterinarians and producers are implementing AMS as part of routine animal health management programs.

Metritis is the second-most treated disease of dairy cows, and systemically administered ceftiofur is the most used compound for the treatment of that condition. Ampicillin is an approved alternative to ceftiofur, but the use of aminopenicillins may result in selection for AMP-C-producing isolates, which are also of concern for human health [37]. A recent review described the complexities of defining appropriate treatments for metritis and called for additional studies to identify which cows benefit from antimicrobial treatments [38]. While modifying treatment protocols is an important aspect of AMS, with the limited number of therapeutic options, veterinarians must focus on prevention. When treatment protocols include watchful waiting (observation of an affected animal without administration of antimicrobials), veterinarians must ensure that producers have recording and diagnostic capabilities to monitor the animals and to ensure animal well-being.

Reducing AMU has often been the presumed objective of AMS, but antimicrobial usage varies considerably among farms and “normal” usage is not well defined. Most farmers and their veterinarians do not have easy access to metrics describing AMU. Even when data are available, it is difficult to compare AMU among studies and regions as a variety of metrics and methodologies have been used to quantify it [14,39,40,41,42,43,44,45]. While each metric has advantages and disadvantages, the use of defined daily doses [46] (DDD) or animal defined doses has been reported the most frequently [47,48,49,50,51]. Despite differing methods of collecting and summarizing data, variability in herd sizes, and differences in distribution and availability of products, there is surprising consistency in the estimations of AMU used in mature cows on dairy farms (Table 2).

Most researchers have reported an AMU of about 5–7 DDD per cow per year, but some researchers have reported less. Using a mailed survey that asked farmers to recall treatments, researchers estimated usage to be about 1.5 ADD/animal/year (youngstock were included in the denominator) for 233 small dairy farms in PA [45]. However, only 63% of farmers reported having written treatment records, so recall bias may have influenced these results, illustrating that the first step to initiating AMS on dairy farms is ensuring that adequate recording systems are used.

While the average values for AMU on dairy farms are surprisingly similar, considerable variation in AMU among farms has been consistently described. Stevens et al. (2016) collected discarded drug packaging to estimate AMU on 57 dairy farms in Belgium and reported that it ranged from 8.7 to 42 DDD per cow/year among farms. Tremendous variability in AMU has been documented among large Wisconsin dairy farms enrolled in a study based on entering most treatments in electronic dairy management records [14]. Combined AMU in preweaned calves and mature cows ranged from 6 to 43 defined daily doses per 1000 animal days (2–16 DDD per cow and preweaned calf/year), [14]. Of the 35 farms that raised preweaned calves on-site, AMU in preweaned calves was highly variable, ranging from 0.3 to 135.4 ADD/1000 preweaned calf days. For mature cows, AMU ranged from about 2 to 12 DDD per cow per year with most antibiotics used for the treatment of mastitis or at dry off (Figure 1). This variability in AMU indicates that some farms have opportunities to reduce AMU by identifying drivers for treatments on their farms. In these instances, the reassessment of criteria for disease detection and recording, risk factors contributing to the incidence of disease and a review of compliance with treatment protocols is an important aspect of AMS for local veterinarians.

## 3. Challenges of Implementing Antimicrobial Stewardship in Dairy Practice

North American veterinarians are highly engaged with most dairy producers. In a nationally representative survey, only about 6% of U.S. dairy farmers indicated that they do not use veterinarians and veterinarians regularly visit most dairy farms on a weekly or monthly basis [52]. However, the health management of dairy cattle on U.S. farms is complex and relies on farm workers for disease detection, diagnosis and administration of primary treatments, while veterinarians are responsible for providing an oversight [53]. Among smaller conventional and organic farms enrolled in a multistate study (n = 292 farms), except for the diagnosis of metritis, <10% of farms reported using veterinarians to initially diagnose diseases of dairy cattle and less than half of diseased animals had been examined by a veterinarian [54]. Similar trends have recently been reported for large dairy farms in the western U.S., where veterinary input was included in less than half of decisions about the use of injectable antibiotics for the treatment of sick cows [55]. In a nationally representative survey, disease diagnosis/treatment or milk and meat drug residue avoidance were listed among the “top three” veterinary services by 45% and 5% of producers, respectively [52]. Veterinarians are most engaged in reproductive management and only about half of U.S. dairy farmers who milked > 500 cows reported that their treatment protocols were designed by the herd veterinarian [52]. The disconnect between the provision of primary care and oversight of treatment protocols creates difficulties for veterinarians seeking to assess the appropriateness of AMU on individual farms. While veterinarians provide prescriptions, drug purchases on larger farms frequently go through distributors. While the majority of small- and medium-sized dairy producers purchase prescription drugs directly from their local veterinarian, the majority of larger producers purchase drugs from distributors based on prescriptions mailed or delivered by the veterinarian [52]. On an economic basis, it is difficult to justify greater engagement of veterinarians in providing primary care to dairy cattle. While mastitis is the most common disease of dairy cows and most frequent reason for AMU [14,17,47,48], most cases present with mild or moderate clinical signs that are routinely diagnosed and treated by farm workers [13].

Veterinarians are considered an important source of advice about AMU [56] and are frequently consulted, but most N. American dairy producers rely heavily upon their own previous experience to guide treatment decisions [11,13] and often consult veterinarians only for cases that require extra-label treatments [7]. As most producers perceive that they are using the “right amount of antibiotics” [6] and have few benchmarks that define antimicrobial usage [53], there is little motivation for dairy farmers to modify their treatment practices. These issues are recognized by veterinarians. Dairy veterinarians who participated in focus groups noted that the ability to communicate with farm decision makers, difficulties in monitoring AMU, and the inability to demonstrate value were all barriers to the implementation of AMS on dairy farms [53]. Similar barriers were identified in a scoping review that also noted that reduced AMU is an outcome that may not be meaningful to farmers who tend to be focused on maintaining the welfare and productivity of their animals [57]. Rather than emphasizing a consumer-driven need to reduce AMU, reframing AMS as a mechanism to reduce costs and improve animal well-being may be provide more motivation and align better with producers’ priorities.

The implementation of AMS requires increased engagement of veterinarians with producers, both of whom are ambivalent about the need and potential benefits of AMS [58]. Despite this considerable ambivalence, the large range in AMU among farms (Figure 1) indicates that farmers who use fewer antimicrobials have recognized the benefits of preventing bacterial diseases, often without explicitly engaging in an “AMS plan”. The ability to engage farmers who use greater amounts of antimicrobials is dependent on demonstrating that their current level of AMU is not “normal” and convincing them of economic and animal welfare benefits that will accrue based on a reduced need for AMU if bacterial diseases are prevented and evidence-based treatment protocols are used.

The ability to assess AMU is one principle of stewardship, but few dairy farmers can describe how AMU on their farms compares to peers and few veterinarians in N. America have ready access to data describing AMU for their agricultural clients. European researchers have reported that farmers take great pride in ensuring the health of their animals [59] and AMU is motivated by a desire to provide good animal care [60]. We previously surveyed almost 600 Wisconsin dairy producers and reported that >90% of those using antibiotics believed that they were using the “right amount” of antibiotics on their farms [6]. The American Veterinary Medical Association recently issued a call to action for veterinarians to collect data on AMU [61], but this is logistically challenging and very time consuming. Like other management areas, it is difficult to motivate change in the absence of measurements and feedback. Benchmarking is a mechanism that is commonly used by veterinarians to assess reproductive performance and health management on dairy farms and is based on comparisons of specific performance metrics to peers. In contrast to benchmarks used for other management areas (such as reproductive performance, somatic cell count, etc.), the benchmarks for assessing AMU are rarely available and have not been standardized. Most dairy farmers do not know if AMU on their farms differs from peers, nor if it is excessive and should be reduced. While several European countries have adopted benchmarking as part of their AMS programs [62], animal health records on U.S. dairy farms are privately held and the lack of a centralized database for U.S. dairy farmers has been a considerable barrier. Benchmarking AMU for individual farms requires a commitment to accurately recording treatments and facilitates measurements of compliance with treatment protocols. The ability to generate benchmarks for AMU on U.S. dairy farms is limited by access to reliable records, difficulties in standardizing outputs, and a limited database for comparisons. Some tools are beginning to be available (https://dairyantibioticbenchmark.msu.edu/, accessed on 8 August 2022), but broader comparison groups that include validated data are needed.

Defining appropriate AMU on dairy farms may motivate some farmers to engage in AMS but changing the behavior of individuals is a complex process that includes both external and internal motivators. The difficulty in changing behaviors associated with AMU in veterinary medicine has been noted by researchers who have used techniques from social sciences to find ways to motivate farmers and veterinarians who contribute to this complex animal health system [63,64]. In the absence of governmental or purchaser mandates to reduce AMU, gradual changes should be expected as U.S. veterinarians and producers increasingly recognize the benefits of disease control.

## 4. Defining and Implementing Antimicrobial Stewardship on Dairy Farms

The American Association of Bovine Practitioners has guidelines for implementing AMS and defines it as a “Commitment to reducing the need for antimicrobial drugs by preventing infectious disease in cattle, and when antimicrobial drugs are needed, to using antimicrobial drugs appropriately to optimize health outcomes as well as prevent violative residues” [65]. The guidelines are specific to bovine practice and acknowledge that bovine practitioners share responsibilities and should engage all stakeholders who influence AMU on farms. The key elements of AMS in bovine practice include leadership in managing the cycle of bacterial disease, drug expertise (including the provision of farm-specific treatment protocols), tracking and benchmarking AMU, and educating key farm workers. The prevention of bacterial disease is emphasized, but to broadly implement AMS on dairy farms, veterinarians must be able to effectively communicate benefits (beyond an expected reduction in AMU) to farm decision makers. For this to work, veterinarians need access to benchmarks and better tools for monitoring the outcomes of changes attributable to AMS.

Historically, AMS in dairy practice has emphasized the reduction in residues in meat and milk [66], but until recently little emphasis has been placed on reducing risks associated with AMU selecting for resistant pathogens. Most U.S. dairy producers participate in an industry-sponsored quality assurance program (https://nationaldairyfarm.com/, accessed on 8 August 2022) that includes a module on antibiotic stewardship that emphasizes the prevention of drug residues but is gradually introducing concepts related to curbing the emergence of AMR. Few treatment failures on dairy farms have been attributable to resistance, so preventing disease and reducing treatment costs may be more motivating to some producers.

The key elements and structure of AMS have been broadly defined [10,57,65,67,68], but a gap remains between acknowledging a need for AMS and the ability to convince dairy farmers that devoting resources to AMS will benefit their businesses. Weese et al. (2013) defined the “5-R principles of stewardship” for veterinary medicine, which include the concepts of Responsibility, Reduction, Replacement, Refinement and Review. These concepts align well with the principles of AMS described for bovine practitioners [65] and provide opportunities for veterinarians to engage with dairy farmers to prevent bacterial disease while improving animal well-being and potentially reducing costs. Most of the principles of AMS described in these documents are aligned with preventive health management practices that are regularly performed by dairy practitioners as part of their health management services (Table 3). Normalizing AMS as a standard aspect of dairy practice requires recognizing that many practitioners have long engaged in practices focused on reducing the need for antimicrobials. It is difficult to track the impact of these practices but the steady decline in morbidity rates for most diseases of dairy cattle illustrates the positive impact that has already occurred. Reframing AMS to acknowledge existing efforts and emphasize benefits rather than obligations may be helpful to overcome the ambivalence of both dairy producers and veterinary practitioners.

## 5. Initiating Antimicrobial Stewardship on Dairy Farms

With or without the appropriate label, most dairy practitioners are engaged in elements of AMS on many dairy farms and increased involvement can be achieved in a stepwise manner. Dairy cattle are at the greatest risk for bacterial diseases at distinct periods, such as the preweaning period for calves and the transition period for adult cows. Reviewing and revising the treatment protocols and preventive practices for mastitis, metritis, and respiratory disease in lactating cows and diarrhea and respiratory disease in calves are obvious steps that can immediately impact both AMU and reduce morbidity. A stepwise process to systematically review the detection, diagnostic criteria, treatment protocols and outcomes of a single high-priority disease can be integrated into scheduled herd-visits, allowing the process to become a routine part of each visit consuming excessive management time (Figure 2).

It is not possible to manage disease without knowing which animals are affected and, in the first month, veterinary practitioners should assess methods of defining, detecting, and recording disease. For example, for mastitis control, it is necessary to ensure that foremilk is examined to detect mild signs of clinical mastitis and indirect tests of inflammation (such as SCC) must be regularly performed to detect subclinical cases. Most importantly, diseases must be recorded in permanent records that are actively used to determine new infection rates as well as compliance with treatment protocols. As antimicrobials are indicated only for specific bacterial infections, knowledge of etiology is fundamental to AMS and, when possible, should be incorporated into treatment protocols for clinical cases. In month 2, the next step is to determine etiology by performing appropriate diagnostic testing. Simple methods of bacterial culture are often used to identify mastitis pathogens that may be responsible for approved antimicrobials [69]. As the AMS cycle progresses, compliance with veterinary prescribed protocols should be assessed. In many herds, slight modifications to mastitis treatment protocols (changes in duration or substitutions of intramammary products) are often made without veterinary consultation, and including a routine review of compliance in herd visits may enhance the engagement of veterinarians with farm workers who deliver treatments. Finally, in the fourth month, the efficacy of the outcomes of treatments can be reviewed, with an emphasis on determining if all treatments are necessary. While some outcomes of mastitis therapy are difficult to assess on farms [30], post-treatment outcomes such as retention within the herd or SCC trends after treatment can be useful to help determine which future cases may benefit from antimicrobial therapy.

This process can be repeated for other high-priority diseases and become a standard part of routine herd visits. Many dairy veterinarians have routine bi-weekly herd visits (eight visits during a 4-month period) and can apportion tasks within the scheduled visits to meet time constraints that are inherent to dairy practice. A four-month cycle allows veterinarians to address three high-priority diseases per year and ensure continuous improvement by including AMS as part of routine expectations during herd visits.

## 6. Conclusions

Antimicrobial usage in food-producing animals is increasingly scrutinized based on concerns about the development and dissemination of resistant bacteria that threaten human health. The U.S. dairy industry accounts for a relatively small proportion of overall antimicrobial usage but shares responsibly to reduce the usage of the highest-priority, critically important antimicrobials. Consumers and some regulatory agencies are skeptical of the ability of farmers and veterinarians to appropriately manage antimicrobials while agricultural stakeholders have considerable ambivalence about the need and ability to reduce antimicrobial usage. Antimicrobial stewardship includes a defined set of processes that focus on reducing the need for antimicrobials through the prevention of bacterial diseases and ensuring that when animals become infected, evidence-based treatment protocols are used. Barriers to AMS include difficulties in benchmarking AMU, unrecognized benefits for farmers and a lack of consistency in disease detection, diagnosis, and recording. These barriers can be overcome by proactively including AMS principles in routinely scheduled herd visits. Many principles of AMS are already embedded within preventive health programs commonly practiced by dairy veterinarians, but there is an opportunity to proactively engage with producers to ensure that benefits attributable to AMS are recognized and embraced by the dairy industry.

## Figures and Tables

**Figure 1 microorganisms-10-01626-f001:**
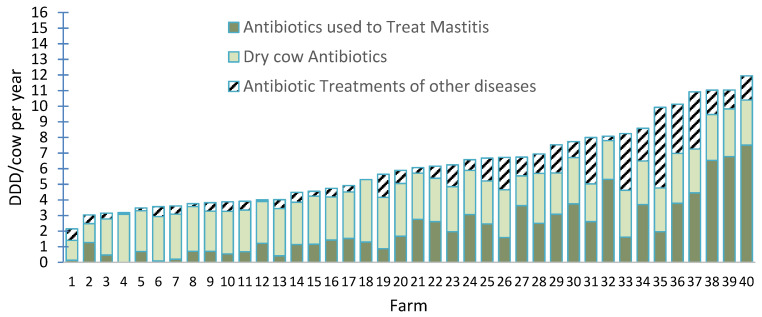
Antimicrobial usage estimated as defined daily doses (DDD) per cow per year on 40 large WI dairy farms in 2017 by indication. Adapted from [14].

**Figure 2 microorganisms-10-01626-f002:**
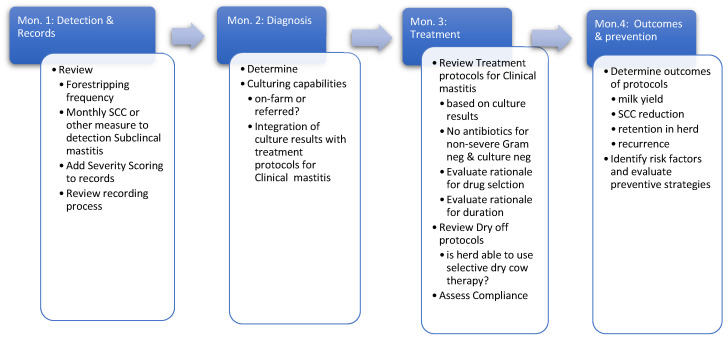
Proposed 4-month cycle integrating principles of antimicrobial stewardship into dairy practice using management of mastitis as an example.

**Table 1 microorganisms-10-01626-t001:** Active ingredients of antimicrobials approved for the treatment of dairy cattle in the United States.

Usage	Active Ingredient of Products Approved for Specific Indications Highest Priority, Critically Important Antimicrobials [15]	
Intramammary treatment and control of mastitis	Lactating cows—7 products Ceftiofur; Cephapirin; Amoxicillin; Hetacillin; Pirlimycin; Cloxacillin; Penicillin	Dry cows—7 products Ceftiofur; Cephapirin; Penicillin/Streptomycin; Penicillin; Albacillin; Penicillin/Novobiocin; Cloxacillin
Systemic ^a,b^ treatments	Cattle ≥ 20 months of age—7 approved formulations of 5 antimicrobials Ampicillin; Ceftiofur (3 formulations); Oxytetracycline. Penicillin and Sulfadimethoxine (restricted usage)	
Calves ^a,b^ and replacement heifers	Calves and cattle < 20 months of age—10 approved antimicrobials Ampicillin; Ceftiofur, Enrofloxacin; Florfenicol; Gamithromycin; Oxytetracycline; Penicillin; Sulfadimethoxine (restricted use); Tildipirosin; Tulathromycin	

Notes: ^a^ Additional antimicrobials (e.g., lincomycin, spectinomycin, gentamycin, and tylosin) may occasionally be prescribed for extra label usage by an attending veterinarian when no labeled products are expected to be efficacious, but extended withholding periods are often required; ^b^ most products are labeled for the treatment of bovine respiratory disease and/or foot rot.

**Table 2 microorganisms-10-01626-t002:** Comparison of antimicrobial usage in mature dairy cows among selected studies that reported using defined daily doses (converted to defined daily doses per cow per year).

	Pol & Ruegg [47]	Saini et al. [48]	Gonzalez Pereyra et al. [49]	Kuipers et al. [50]	Stevens et al. [51]	de Campos et al. [14]
Year	2007	2012	2015	2016	2016	2021
Country	WI, USA	Canada	Argentina	Netherlands	Belgium	WI USA
Data Collection method	On-farm survey	Packaging audit	On-farm survey	Retrieval of sales data	Packaging audit	Farm visit and record analysis
Herd (n)	40 ^1^	89	18	94	57	40
Lactating Cows/herd	197	69	219	110	69	1163
DDD/cow/year	5.4	5.2	5.2	5.5	7.6	6.1
% IMM route ^2^	66%	35%	85%	72%	63%	78%

NOTES: ^1^ 20 conventional and 20 organic herds; ^2^ IMM = intramammary route of administration.

**Table 3 microorganisms-10-01626-t003:** Principles and elements of AMS that are part of routine veterinary practice on many dairy farms.

		Element of AMS	
		AABP, 2022 [66]	Weese et al., 2013 [68]
Core Principle of Judicious Use ^1^	Examples of Common Practices Related to Antimicrobial Stewardship	Leadership; Drug Expertise; Tracking AMU; Reporting and Action	Responsibility; Reduction; Replacement; Refinement; Review
**Prevention**	Perform housing assessment Reduce stocking density Modify Gram-negative core antigen vaccine Modify fresh cow environment Change milking order Evaluate and change colostrum harvest protocol Test for failure of passive transfer Evaluate calf and heifer housing Evaluate heat stress control Begin daily herd walk-through Evaluate and revise biosecurity programs Evaluate and revise vaccination programs Review use of shared versus individual needles	**ACTION** “Review the disease prevention programs such as vaccination, nutrition, and environmental management programs for specific disease conditions to assure optimal husbandry.”	**REDUCTION** “…requires consideration of the entire spectrum of possible reduction approaches, which also include genetic selection for disease resistance, use of vaccines…, identifying modifiable risk factors and of course, measuring current practice.”
**Diagnosis**	Begin monthly SCC testing Use CMT at dry off to identify infected quarters Use SCC as part of clinical mastitis treatment protocol Use SCC as part of selective dry cow therapy program Review disease definitions Use on-farm culture to direct clinical mastitis treatment protocols Begin calf health scoring Recording severity score for clinical mastitis Begin fore-stripping (detection of clinical mastitis)	**ACTION** “Review diagnosis/treatment protocols developed for different disease syndromes.”	**REFINEMENT** “… improved culture-based diagnostic tests are allowing selective treatment of dairy cattle with purulent vaginal discharge or clinical mastitis, and improved use of SCC data is guiding selective dry cow treatment of dairy cattle, each decreasing amu.”
**Drug Selection and Management**	Use “watchful waiting” for some non-severe mastitis Evaluate selective Dry cow therapy Evaluate use of internal teat sealants Develop drug inventory control program Requires 2 signatures for culling cows Change protocol for cows to leave hospital/sell milk	**DRUG EXPERTISE** “Bovine practitioners should provide AMU protocols and treatment guidelines specific for each operation as described in the AABP Guideline” “Establishing and maintaining the veterinarian-client-patient relationship in bovine practice” and “Drug use guidelines for bovine practice.”	**REFINE and REPLACEMENT** “Replacement of the use of antimicrobials with alternative, nonantimicrobial measures, wherever possible and appropriate, is another critical AMS tenet”
**Treatment Protocols**	Plan a selective dry cow therapy program Review (change) routine fresh cow treatments Review/revise existing treatment protocols Reduce use of extralabel treatments Review reasons for using antibiotics Review evidence for duration of antibiotic therapy Weigh cattle before dosing injectable antibiotics Review or change criteria for use of antibiotics Review or change antibiotics used for treatment Compare cost of treatment protocols Read labels of all drugs on farm Identify compliance gaps for treatment protocols Schedule quarterly review of treatment protocols Schedule training time of vet with animal health managers	**LEADERSHIP** “Am I committed to complete the cycle of disease management by following the judicious use of antimicrobial drugs with reevaluation of their need?” “Have I followed the legal requirements for using antimicrobial drugs by selecting approved products when available or choosing legally acceptable extra-label use?”	**REFINEMENT** “Use all antimicrobial agents, … only after careful review and reasonable justification.” “Use narrow-spectrum in preference to broad-spectrum antimicrobials whenever appropriate.” “Minimize therapeutic exposure to antimicrobials by treating for only as long as needed to meet the therapeutic objective.”
**Monitoring**	Perform residue tests on milk of fresh cows Evaluate transition cow health Change disease recording system Review health records to estimate disease rates Review calf health records Conduct quality assurance review of on-farm culture results Evaluate outcomes of mastitis treatment program Review cost of drugs Review disease rates in calves/heifers Determine death and culling rates for animals Benchmark AMU	**TRACKING and REPORTING** “Bovine practitioners should periodically review treatment records, drugs present on the farm in relation to treatment protocols, and on-farm antimicrobial drug dispensing and usage.” “Bovine practitioners should support efforts to report AMU across farms in order to benchmark and compare usage...”	**REVIEW** “Review includes the measurement of progress toward each objective. Information on the use of antimicrobials can be obtained from both quantitative and qualitative assessments.”
**Leadership**	Assign fresh cow manager Provide training for on-farm culture program Change personnel making treatment decision Limit access to antibiotics Have veterinarian reassess/review current VFD Begin routine training program for employees	**LEADERSHIP** “It includes accepting responsibility and accountability for antimicrobial prescribing, dispensing, and administration. This commitment also includes identifying leaders within the practice and client operations to share in antimicrobial stewardship.”	**RESPONSIBILITY** “There are many enabling mechanisms to ensure that a collaborative and participatory team approach is taken with effective communication with all stakeholders.”

NOTES: ^1^ adapted from [68].

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
