# Peer review of "Realities, Challenges and Benefits of Antimicrobial Stewardship in Dairy Practice in the United States"

_microorganisms, 2022, doi:10.3390/microorganisms10081626_

Round 1

Reviewer 1 Report

The article is very interesting, but it's about treating cows mainly in the USA. 

The introduction of antibiotics into clinical practice represented one of the most important interventions for the control of infectious diseases. Antibiotics have saved millions  of human and animal lives  and have also brought a revolution in medicine and veterinary medicine.

The global emergence of antimicrobial resistance has become a preeminent concern in medicine,  public health and veterinary medicine.   Dramatic overuse and misuse of antimicrobial agents around the world must be reduced.

Bovine mastitis plays a decisive role in the dairy industry and causing considerable economic losses. Emergence and spread of antimicrobial resistance  is an urgent matter of particular public interest, and as a consequence the article is very important.

Similar articles are from 2008-2017

 Krömker V, Leimbach S. Mastitis treatment-Reduction in antibiotic usage in dairy cows. Reprod Domest Anim. 2017 Aug;52 Suppl 3:21-29. doi: 10.1111/rda.13032. PMID: 28815847.

Oliver SP, Murinda SE. Antimicrobial resistance of mastitis pathogens. Vet Clin North Am Food Anim Pract. 2012 Jul;28(2):165-85. doi: 10.1016/j.cvfa.2012.03.005. Epub 2012 Apr 28. PMID: 22664201.

Call DR, Davis MA, Sawant AA. Antimicrobial resistance in beef and dairy cattle production. Anim Health Res Rev. 2008 Dec;9(2):159-67. doi: 10.1017/S1466252308001515. Epub 2008 Nov 5. PMID: 18983724.

Author Response

Thank you for the time that you spent reviewing this article, I appreciate your comments.  I agree that the manuscript is focused on the U.S. situation and have pointed that out on lines 68-70.  As regulations and veterinary practice culture vary among countries, I thought it was best to focus on the N. American situations. 

Reviewer 2 Report

The manuscript summarized and discussed antimicrobial stewardship programs for dairy practice. Plenty of information was provided by the author, which is of interest to the readership. Please consider the following aspects to further improve the manuscript.

1. The overall structure looks fine, but it would be better to have one topic/focus in each paragraph. For example, the Introduction can be divided into 2-3 paragraphs. Same suggestion for the 2nd paragraph of Section 2 and 1st paragraph of Section 3.

2. Add “Note:” to the footnotes of tables; otherwise, the current notes are not well distinguished from the main text.

3. As an enlightening direction, Section 5 needs to be expanded, particularly regarding Figure 2 using mastitis as an example.

Author Response

Thank you for the time that you spent reviewing this article.  I appreciate your input as peer review always improves the quality of manuscripts.  I have attempted to fully respond to your comments.

  1. The overall structure looks fine, but it would be better to have one topic/focus in each paragraph. For example, the Introduction can be divided into 2-3 paragraphs. Same suggestion for the 2nd paragraph of Section 2 and 1st paragraph of Section
    1. AU:  I added separations in paragraphs as suggested.  As I did not originally indent paragraphs, I think my existing separations were not always evident, so I have also tried to indent all.  Thank you for this suggestion. 
    2. Add “Note:” to the footnotes of tables; otherwise, the current notes are not well distinguished from the main text.
      1. AU:  Done
    3. As an enlightening direction, Section 5 needs to be expanded, particularly regarding Figure 2 using mastitis as an example.
      1. AU:  I considerably expanded my description of Figure 2.  Thank you for that suggestion.  The added text can be found on lines 335-361.

Reviewer 3 Report

The manuscript ID microorganisms-1813600 entitled " Realities, Challenges and Benefits of Antimicrobial Stewardship in Dairy Practice " by Pamela L Ruegg to define elements of Antimicrobial Stewardship programs for dairy practice and discuss challenges and potential benefits associated with Antimicrobial Stewardship concepts by this review. It was very important point of the dairy industry among food-producing animals’ industry, and this review will be fit into the Special Issue "Antimicrobial Stewardship in Food-Producing Animals 2.0". However, there are a few points to be revised in this manuscript.

Author Response

Thank you for the time that you spent reviewing this article, I appreciate your insights and have attempted to respond to all of them.  Here are my responses:

  1. This contents of review from U.S. not worldwide. Therefore, this title may be overstating.You had better to title "Realities, Challenges and Benefits of Antimicrobial Stewardship in
    Dairy Practice in U.S."
    1. This is a good suggestion and I made this change.
  2. P1L27: Your keyword was poor, and you should add “antimicrobial resistance”.
    1. Done
  3. P1L20: in human medicine?
    1. I wasn't sure where this comment referred to and was unable to make any changes.
  4. The layout and design of these Table and Fig was not fit in the manuscript and the readers can read and understand very hard.
    1. I will work with the editorial staff of the journal to ensure that all tables fit well and are understandable.
  5. IMM route? What is it?
    1. thank you for finding this omission.  I added a footnote to the table to define IMM as "intramammary route"
  6. P5L197: PWC-d
    1. Again - thanks for finding this oversight, I wrote out pre-weaned calf days.